# Synthesis of fluorescent organic nano-dots and their application as efficient color conversion layers

Yeasin Khan [1], Soonjae Hwang [2], Ramanaskanda Braveenth[2], Young Hun Jung[2], Bright Walker [1✉] & Jang Hyuk Kwon [2✉]

Efficient conversion of light from short wavelengths to longer wavelengths using color conversion layers (CCLs) underpins the successful operation of numerous contemporary display and lighting technologies. Inorganic quantum dots, based on CdSe or InP, for example, have received much attention in this context, however, suffer from instability and toxic cadmium or phosphine chemistry. Organic nanoparticles (NPs), though less often studied, are capable of very competitive performance, including outstanding stability and water-processability. Surfactants, which are critical in stabilizing many types of nano-structures, have not yet been used extensively in organic NPs. Here we show the utility of surfactants in the synthesis and processing of organic NPs by thoroughly characterizing the effect of ionic and non-ionic surfactants on the properties of fluorescent organic NPs. Using this information, we identify surfactant processing conditions that result in nearly 100 % conversion of organic fluorophores into sub-micrometer particles, or nano-dots, with outstanding performance as CCLs. Such water dispersions are environmentally benign and efficiently convert light. They can be used for a range of fluorophores covering a full spectral gamut, with excellent color purity, including full-width at half-maximum (FWHM) values as low as 21 nm. Compared to inorganic (InP) reference CCLs, the organic nano-dot based CCLs show superior color conversion efficiency and substantially improved long-term stability.

[1] Department of Chemistry, Kyung Hee University, 26 Kyungheedaero, Dongdaemun-gu, Seoul, Republic of Korea. [2] Organic Optoelectronic Device Lab. (OODL), Department of Information Display, Kyung Hee University, 26, Kyungheedae-ro, Dongdaemun-gu, Seoul, Republic of Korea. ✉email: walker@khu.ac.kr; jhkwon@khu.ac.kr

Fluorescent organic NPs have attracted tremendous attention worldwide in recent years due to their unique optical and chemical properties that differ from their bulk state[1]. Salient features like facile synthesis[2], easy processability[3], and excellent biocompatibility[4] give them tremendous potential in a wide range of applications including drug delivery[5], biosensing[6], optical sensing[7], cell imaging[8], and so-on, where their inorganic counterparts have been ineffective or incompatible due to their toxicity, non-degradability or physical properties. A wide range of approaches to synthesize organic NPs have been developed; for instance, emulsification[9], nano-precipitation[10], and solvent evaporation[11] are examples of successful approaches that have been reported by many different research groups.

Surface active agents, such as oleate salts and alkyl phosphines, are used extensively in the synthesis of inorganic NPs like Ag, Cu, PbS, ZnS, etc. both in aqueous and non-aqueous dispersions[12,13], however, the use of surfactants in organic NP synthesis has so far been sporadic. Among the most frequently used approaches to synthesize organic NPs is the re-precipitation technique, which involves dissolving organic fluorophores in a solvent, and injecting this solution into a non-solvent to yield nano-crystalline particles[14–16]. Despite impressive NP growth dynamics, this method suffers from poor uniformity and low yields of small particles. As with inorganic NPs, the use of surfactants can greatly improve the uniformity and yield of organic NPs by reducing their surface energy and making NPs the most thermodynamically favorable state. Some reports have used surfactants in the synthesis of organic NPs; for example, cholesterol NPs have been prepared using bis(2-ethylhexyl)sodium sulfosuccinate, polyethylene glycol tert-octylphenyl ether (Triton X-100), and cetyltrimethylammonium bromide as surfactants[17]. In another study, NPs of fluorescent organic materials like 2-(anthracene-9-yl)-9,9-dioctyl-fluorene and 2-(anthracene-9-yl)-spirobifluorene were synthesized by semi-continuous emulsion polymerization of MMA and a polymerizable surfactant sodium 3-(allyloxy)-2-hydroxypropane-1-sulfonate[18]. Aggregation-induced emission-based fluorescent organic nanoparticles were also prepared in another study using a commercial surfactant namely, poly(-ethyleneglycol)-block-poly(propyleneglycol)-block-poly(-ethyleneglycol) diacrylate (F127) for cell imaging purposes[19]. Due to concentration quenching[20,21] effects observed in many fluorescent molecules, NPs based on molecular fluorophores have not been investigated as often as inorganic NPs, and systematic, quantitative analysis of the effects of surfactants on organic NP properties such as yield, morphology, and photophysical properties, as well as the influence of surfactant properties, such as CMC, is currently lacking.

Fluorescent and phosphorescent organic dyes have immense commercial importance in light emission applications. In recent years, they have evolved to suit new high-tech applications such as organic light-emitting diodes (OLEDs)[22], organic light-emitting transistors (OLETs)[23], and color generation devices as both active materials and color conversion layers (CCLs). Although light-emitting molecules and polymers are well known in these fields, NPs of fluorescent organic materials are comparatively less explored.

Despite being relatively inconspicuous in the literature, organic NPs offer several key advantages compared to other materials, which has motivated us to study their application in optoelectronic devices. We believe that the study of organic NPs in optoelectronics will open new possibilities in the field of light-emitting diodes (LEDs) and displays, especially as CCLs. A variety of phosphor-based materials[24–26] has been used in CCL fabrication to produce white light in the recent past. Apart from phosphors, many luminescent materials like inorganic quantum dots (QDs)[27–29], perovskites[30,31], and organic dyes[32–34] have been incorporated as CCLs in recent reports. Organic dyes, despite showing reasonable performances as CCLs, suffer greatly from aggregation resulting in emission quenching[35] and poor photochemical stability[36]. Inorganic QDs are also very promising candidates for constructing CCLs as they exhibit good photoluminescence quantum yield (PLQY), good color purity, reasonably good stability, and better light scattering effects. However, inorganic QDs are highly sensitive to moisture and temperature[37]. They also involve undesirable synthetic procedures involving toxic heavy metals like arsenic, cadmium, lead, or spontaneously flammable, malodorous reagents like phosphines. In contrast, fluorescent carbon dots and carbonized polymer dots have shown practical importance in white LEDs[38–42]. Similarly, organic NPs also offer the possibility to fabricate CCLs with enhanced photochemical stability, moisture stability, and high uniformity using facile, environmentally-conscientious, synthetic methods.

In this work, we explore the use of both ionic and non-ionic surfactants to synthesize uniform, water-based dispersions of NPs (which we will refer to as nano-dots) of fluorescent organic materials at very high yields. We include a comprehensive study of the effects that surfactants have on the properties of the fluorescent organic nano-dots, which, to our knowledge, has not been reported before. Moreover, we demonstrate the application of the nano-dots as CCLs, which result in high color conversion efficiencies, good photochemical stability, and excellent repeatability. We believe that CCLs prepared using fluorescent organic nano-dots will have wide applicability in light-emitting devices due to their water-processability and exceptional combination of properties compared to conventional CCL materials; they have the potential to give many types of light-emitting devices substantial improvements in stability and performance.

## Results and discussion

**Nano-dot synthesis and characterization.** Surfactants[43] are amphiphilic organic compounds, which contain both hydrophilic and hydrophobic groups. In aqueous solutions, their hydrophobic tails aggregate and self-assemble into cores, while hydrophilic heads remain in contact with the aqueous phase, minimizing interaction between the hydrophobic tails and the polar solution. Such micelle structures spontaneously self-assemble above a certain critical concentration, called the critical micelle concentration (CMC), which is unique to each surfactant. In this study, we have explored the synthesis of organic nano-dots using both ionic and non-ionic surfactants at concentrations below and above the CMC of surfactants to develop an understanding of how surfactants influence the size of organic nano-dots and utilized these nano-dots to make CCLs for LEDs. Nano-dots are formed when a solution of fluorophore rapidly precipitates as it is mixed with a non-solvent (water). If this happens in the presence of a surfactant above its CMC, micelles are able to encapsulate the fluorophore particles as they precipitate, through interaction between the non-polar surfactant tails and non-polar fluorophore molecules. The presence of surfactant molecules is expected to minimize the surface energy of small particles, whereas in the absence of surfactant, the surface energy is minimized when large fluorophore particles are formed; thus small particles (nano-dots) are formed when the precipitation occurs in the presence of a surfactant above its CMC. Since the formation of nano-dots involves a physical change in the state of the fluorophore (no chemical transformation occurs), most of the properties of the fluorophore are not significantly changed, though the absorption and emission spectra of such nano-dot dispersions fall between those of the bulk, solid-state phase and true solution phase of the fluorophore, as can be seen in the photoluminescence (PL) spectra of nano-dots.

In a typical nano-dot preparation, a thermally activated delay fluorescent (TADF) emitter Ttrz-DI (5,10,15-tris(4-(4,6-diphenyl-1,3,5-triazin-2-yl)phenyl)-10,15-dihydro-5H-diindolo[3,2-a:3',2'-c]carbazole) was synthesized and purified (synthetic details can be found in the supplementary information section, Supplementary Figs. 1–4), then dissolved in THF solvent to make a 0.5 mM stock solution. Other fluorophores were used to prepare nano-dots in the same way. Non-ionic Triton X-100 surfactant was dissolved separately in the same solvent to prepare a 0.1 M solution. Then, 0.1 mL of the fluorophore solution was taken in a vial followed by the addition of a variable amounts (0.01, 0.04, 0.1, 0.2, 0.3, 0.4, 0.5 mL) of the surfactant solution. Excess THF was added to keep the total volume constant at 0.6 mL. After that, 4.4 mL of deionized water was injected rapidly into the fluorophore-surfactant solution to make dispersions of nano-dots. The total volume was maintained constant at 5 mL after mixing with non-solvent. The dispersions were filtered through Polytetrafluoroethylene (PTFE) syringe filters with different pore sizes (450 nm and 200 nm diameter) and subjected to dialysis using cellulose acetate tubes for 12 hours to remove the excess surfactant from the dispersions. The dispersions were then concentrated under a reduced pressure of ~0.01 Torr, using a Schlenk line, to evaporate 90% of the solvents. After vacuum evaporation of the excess solvent, the concentrated nano-dot dispersions could then be used or diluted with additional deionized water as necessary to obtain desired concentrations for optical studies and film processing. Figure 1a illustrates the process of synthesizing Ttrz-DI nano-dots using surfactants. The same procedure was followed to synthesize a variety of organic nano-dots with different emission colors using TNAP (N,N,6,10-tetra(naphthalen-2-yl)-6,10-dihydro-6,10-diaza-16b-boraanthra[3,2,1-de]tetracen-8-amine) (material synthesis is provided in supplementary information section, Supplementary Figs 5, 6) 4CzIPN[44] (1,2,3,5-tetrakis(carbazol-9-yl)-4,6-dicyanobenzene, 2,4,5,6-tetrakis(9H-carbazol-9-yl) isophthalonitrile), CzDABNA[45] (2,12-di-tert-butyl-N,N,5,9-tetrakis(4-(tert-butyl)phenyl)-5,9-dihydro-5,9-diaza-13b-boranaphtho[3,2,1-de]anthracen-7-amine), and 4tBuMB[46] (1,3,7,9-tetrakis(4-(tert-butyl)phenyl)-5,5-difluoro-10-(2-methoxyphenyl)-5H-4l4,5l4-dipyrrolo[1,2-c:2',1'-f][1,3,2]diazaborinine) (Fig. 1b–e).

A number of studies have reported using surfactants to synthesize organic nano-dots, however, to our knowledge, a systematic, experimental investigation of the influence of surfactants on the properties of fluorescent organic nano-dots is currently lacking. This motivated us to carry out a comprehensive investigation of the effects of surfactants on the properties of organic nano-dots. We used both ionic and non-ionic surfactants for the synthesis of nano-dots where tetra-butyl ammonium oleate (TBA oleate) (CMC~0.45 mM)[47] represents the ionic surfactant and Triton X-100 represents non-ionic surfactant having (CMC~0.22–0.24 mM)[48].

Concentrations ranging from 0.2 mM to 10 mM of these surfactants were used to synthesize the nano-dots, in order to characterize relationships between surfactant concentration, particle size, and yield of the fluorophore as sub-micron particles. Comparing the PL spectra of the bulk solutions, nano-dot dispersions, and bulk films showed significant differences in peak positions due to the variation in particle sizes. In the solution state, fluorophores exist as isolated molecules, whereas in the bulk film state, fluorophore molecules are aggregated into a bulk, solid structure with dimensions much larger than the size of the molecule. However, in the nano-dot dispersions, the particle size tends to be in micro to nanometer scale with a finite number of molecules in each particle, falling somewhere between bulk solid and solution states. PL data of nano-dot dispersions show peak positions in-between solution and film states as shown in Fig. 2a, b, consistent with the fluorophore existing in an intermediate state between a solid and a true solution.

**Nano-dot size and morphology**. The influence of surfactant concentration on the size and shape of fluorophore particles was initially inspected by optical microscopy (Fig. 2c, d, g and Supplementary Figs. 7–11); surfactant concentration was found to have a dramatic effect on the particle size distribution and morphology. Optical microscopy revealed that without any surfactant, the fluorophore preferentially aggregated into large, continuous solid domains (Fig. 2c, Supplementary Figs. 7, 8) and yielded random sizes and shapes with only a small amount of particles having sizes smaller than 10 μm. However, as the concentration of surfactant was increased, the particle size uniformity increased, whereas the average particle size steadily decreased until reaching a minimum of 120 nm. The same trend was observed for all combinations of fluorophores and surfactants

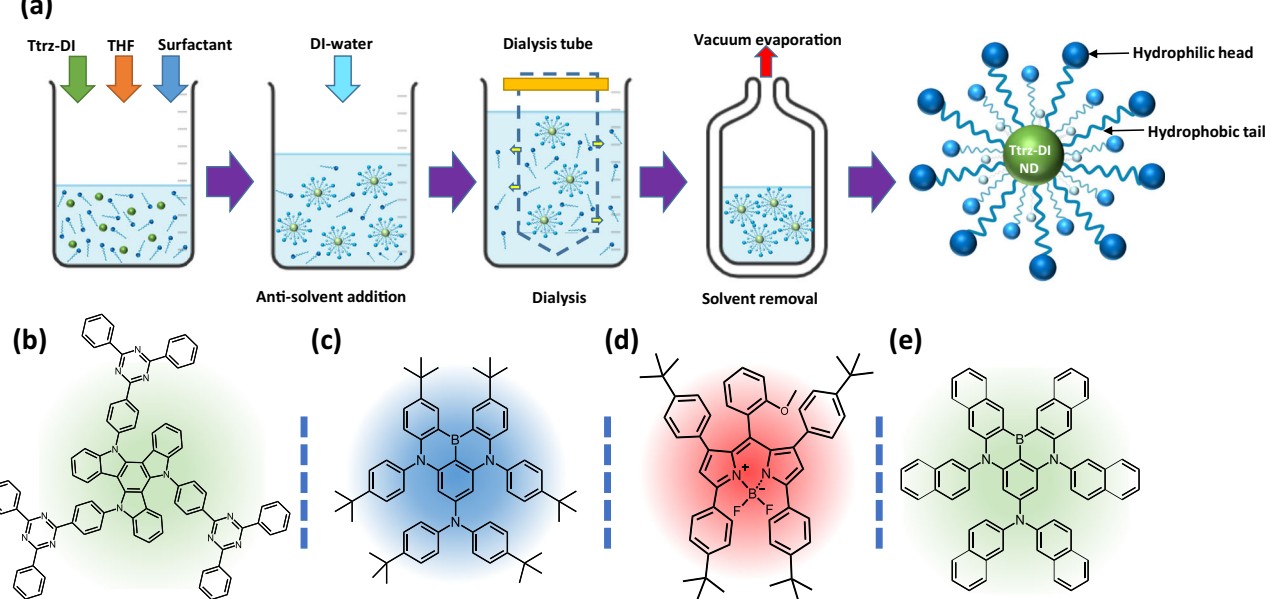

**Fig. 1 Schematic diagrams. a** Synthesis process of Ttrz-DI nano-dot dispersion using surfactants. Chemical structures of **b** Ttrz-DI, **c** CzDABNA, **d** 4tBuMB, **e** TNAP fluorescent materials. **b–e**, the background color corresponds to the emission color of each fluorophore.

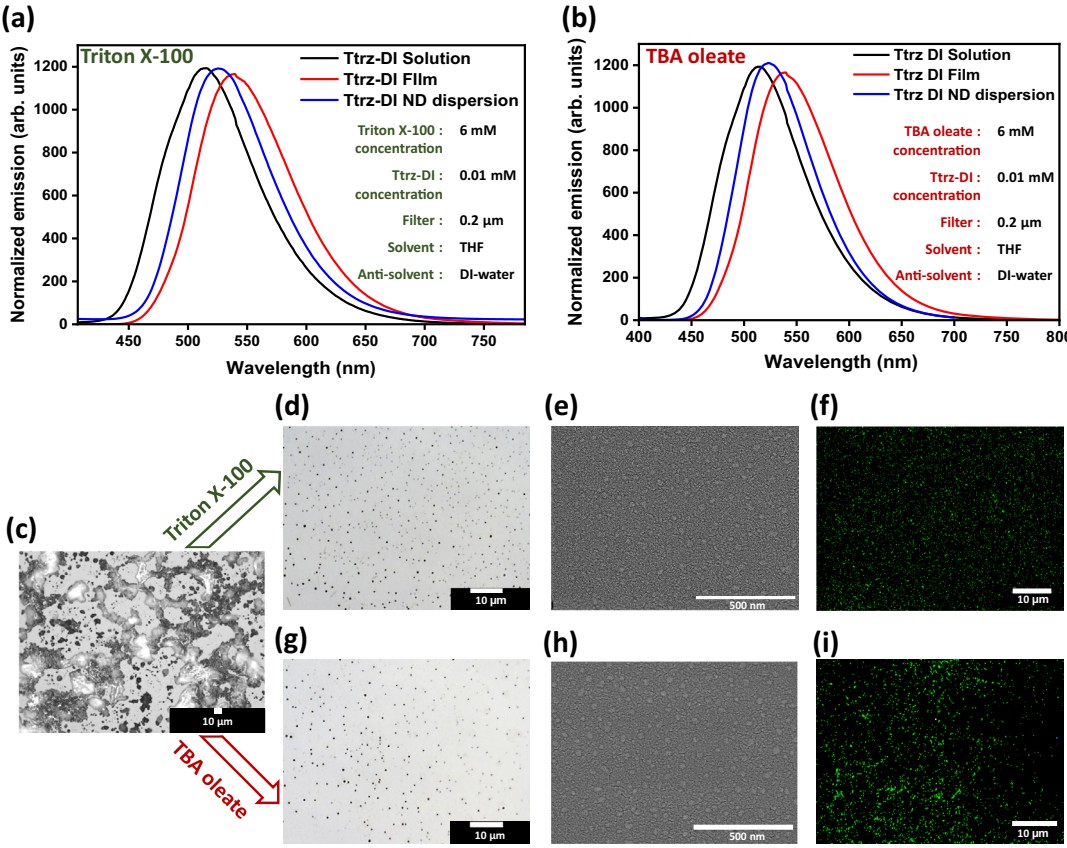

**Fig. 2 PL spectra of solution, film, and nano-dot dispersions of Ttrz-DI.** Prepared using **a** Triton X-100 and **b** TBA oleate. Optical microscope (OM) images of a Ttrz-DI dispersion deposited on a glass substrate synthesized **c** without surfactant and **d** with 6 mM Triton X-100 surfactant. **e** SEM image of a Ttrz-DI ND/Triton X-100 dispersion deposited on a glass substrate. **f** Fluorescence microscope (FM) image of a Ttrz-DI ND/Triton X-100 dispersion deposited on a glass substrate. Optical microscope images of a Ttrz-DI dispersion deposited on glass a substrate synthesized **g** with 6 mM TBA oleate surfactant. **h** SEM image of a Ttrz-DI ND/TBA dispersion deposited on a glass substrate. **i** Fluorescence microscope image of a Ttrz-DI ND/TBA oleate dispersion deposited on a glass substrate.

that were studied. It should be noted that reports of organic nano-dot synthesis by the anti-solvent method predominantly do not involve any surfactant[49], and likely suffer from the inefficient conversion of valuable fluorophore materials into sub-micron particles.

ND dispersions prepared with different surfactant concentrations were studied by fluorimetry. In order to prepare samples for PL measurement, dispersions were filtered through 200 nm PTFE filters to remove any particles larger than 200 nm in diameter. As shown in Fig. 3a, b, and Supplementary Fig. 13, an optimal surfactant concentration was observed, above which a decrease in PL intensity occurred, that coincided with an increase in average particles size. In the case of Ttrz-DI dispersions prepared with Triton X-100, nano-dots started to form when the surfactant concentration was ~0.2 Mm, which incidentally matches the CMC value of Triton X-100. The most intense fluorescence (PLQY 43.4%) was observed at 6 mM surfactant concentration and PL intensity decreased at higher concentrations. A similar trend was observed when Triton X-100 was replaced by the ionic surfactant TBA oleate (Fig. 3a), where fluorescence intensity was generally highest at 6 mM concentration. These data suggest that 6 mM concentration of both ionic and non-ionic surfactants yielded the smallest size of Ttrz-DI nano-dots and the most efficient conversion of the fluorophore to nano-dots. This result was confirmed using several types of microscopy. Atomic force microscopy (AFM), scanning electron microscopy (SEM) and optical microscopy consistently revealed that the smallest average particle size was achieved using 6 mM surfactant concentration.

When other fluorophores were used (4tBuMB and TNAP), the smallest particle size was also obtained at 6 mM surfactant concentration (Supplementary Figs. 10, 11), with the exception of CzDABNA, which showed the smallest average size of 110 nm at 8 mM surfactant concentration. It should be noted that this was only 3 nm smaller than the size (113 nm) obtained using 6 mM surfactant concentration (Supplementary Fig. 9). Therefore, the concentration was fixed at 6 mM for all further experiments with Triton X-100 and TBA oleate. From the AFM analysis, (Supplementary Fig. 14) the deposited particles exhibited flattened shapes, with a significantly smaller height than diameter, indicating that the particles deformed and became flattened on the substrate as the solution dried. The height and width of the deposited nano-dots were used to calculate the volume of the particles on the substrates, and these volumes were used to extrapolate the diameter of the nano-dots as they existed in dispersion, assuming that they had isotropic spherical shapes before being deposited and deforming on the substrates (Supplementary Fig. 15). Dynamic light scattering (DLS) (Supplementary Fig. 16) was also used to measure the particle size in the dispersion, which agreed well with the results of AFM line profile analysis and the aggregation of Ttrz-DI nano-dots overtime was characterized (Supplementary Fig. 17). The particle size was further analyzed by high-resolution SEM, which confirmed the dimensions of the deposited nano-dots. The size was somewhat smaller with SEM than AFM or DLS (Fig. 2e, h, Supplementary Fig. 12 and Supplementary Table 1), likely because the particles were completely dried in the vacuum

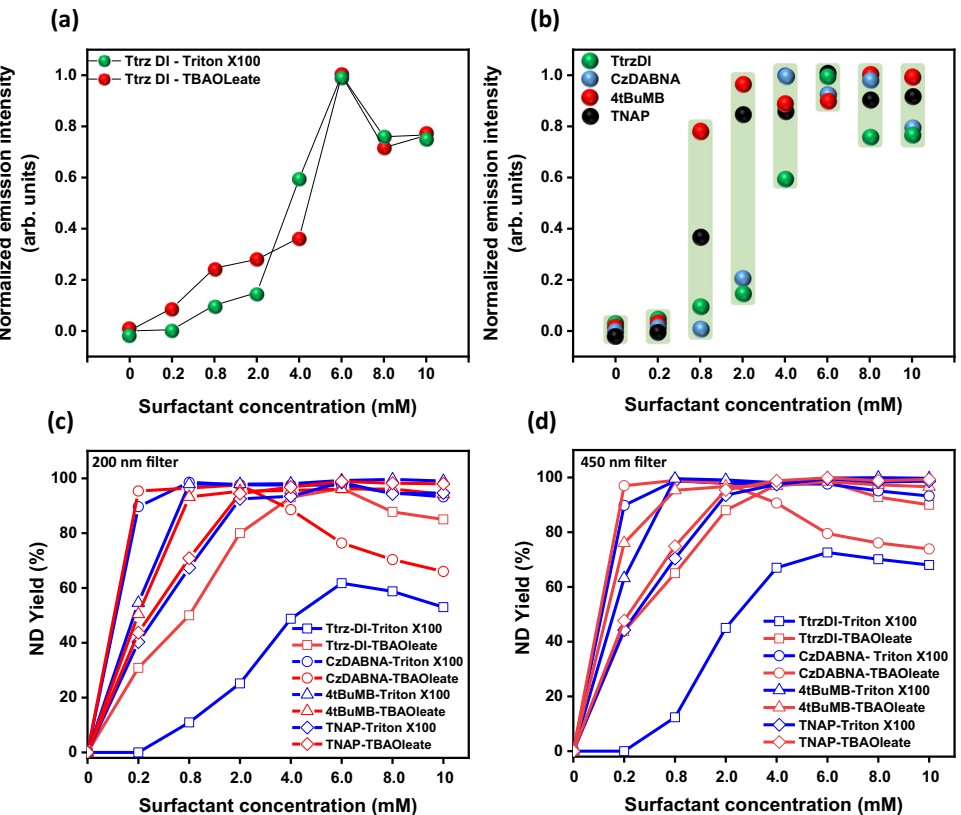

**Fig. 3 PL intensity and yield. a** Relative PL intensity of Ttrz-DI nano-dot dispersions with variable concentrations of Triton X-100 and TBA oleate surfactants, where the concentration of Ttrz-DI fluorophore was kept constant at 0.01 mM. **b** Summary of PL spectra of Ttrz-DI, CzDABNA, TNAP, and 4tBuMB nano-dot dispersions with varyiable concentration of Triton X-100 surfactant. **c** Yield of organic nano-dots of less than 200 nm size at different concentrations of Triton X-100 and TBA oleate. **d** Yield of organic nano-dots of <450 nm size at different concentrations of both Triton X-100 and TBA oleate.

environment during SEM measurement. DLS measures the hydrodynamic radius, which is larger than the true radius of the solid particles due to interactions between the particles and the solvent. Since Ttrz-DI nano-dots are fluorescent in nature, fluorescence microscopy was additionally employed to visualize the particles under 350 nm ultraviolet (UV) illumination (Fig. 2f, i). All of the size measurements consistently revealed the same trends, and similar results were obtained when non-ionic Triton X-100 was replaced with the ionic surfactant TBA oleate (Fig. 2g±i and Supplementary Fig. 8) as well.

**Yield of fluorescent organic nano-dots**. We investigated the yield of Ttrz-DI nano-dots with diameters below a certain threshold (smaller than 200 nm or 450 nm, respectively, based on the amount of material passing through a PTFE filter) using both ionic and non-ionic surfactants. The yield was calculated spectrophotometrically as described in the methods section. The ionic surfactant TBA oleate exhibited a superior yield of Ttrz-DI nano-dots compared to Triton X-100, and the yield of the particles using 6 mM concentration of surfactants was relatively higher than other concentrations. These data show that particles of <200 nm size were synthesized with yields of 61.73% (Triton X-100) and 96.3% (TBA oleate) (Fig. 3c, d). This data also implies that without any surfactant, the non-solvent technique using THF and DI-water mixture produces only a miniscule yield of Ttrz-DI nano-dots, since the average particle size in the dispersion without surfactant was much greater than the filter pore size; consistent with the microscopy results (Fig. 2c–i, Supplementary Figs. 7, 8). Yields of CzDABNA, 4tBuMB, and TNAP were

calculated the same way; Fig. 3c, d summarizes the yields of Ttrz-DI, CzDABNA, 4tBuMB, and TNAP nano-dots of sizes <200 nm and 450 nm, respectively, with different concentrations of Triton X-100 and TBA oleate. Compared with Ttrz-DI, the other fluorophores showed even higher yields, approaching 100%. The yield of nano-dots was largely dependent on the concentration of surfactants relative to the CMC of the surfactants rather than the structure of the fluorophore.

To develop a more complete picture of how different conditions affect nano-dot properties, we explored the effect of using different solvents on the yield of Ttrz-DI nano-dots. The concentration of Ttrz-DI and Triton X-100 was fixed at 0.5 mM and 6 mM respectively with water as the anti-solvent. Apart from THF, other water-miscible solvents including 1,3-dioxalane, 1,4-dioxane, N-methylpyrrolidone (NMP), and N-methyl imidazole were used for the nano-dot synthesis. A slight increase in yield and PLQY was obtained in the case of 1,3-dioxalane and NMP however, the average particle size was found to be the lowest when the solvent medium was THF (Supplementary Fig. 18). Different polar anti-solvents like methanol, ethanol, iso-propanol, and 2-methoxyethanol were also investigated for the ND synthesis. In this case, the concentration of Ttrz-DI and Triton X-100 was kept constant in THF solvent medium. Large-sized visible particles with negligible nano-dot yields were obtained when these anti-solvents were used. Water showed the best yield among all of the investigated anti-solvents (Supplementary Fig. 19). The temperature of the anti-solvent during injection is another important parameter that was investigated. The temperature of the anti-solvent (water) was varied from 25 °C (room temperature) to 90 °C in this experiment. The yield and PLQY of

the nano-dots were found to decrease at increasing anti-solvent injection temperatures and the best yield was obtained at room temperature (Supplementary Fig. 20). Therefore, optimal conditions for nano-dot synthesis included using THF as a solvent and water as the anti-solvent at room temperature.

**Color conversion applications**. Color conversion filters are critical components that make a number of lighting and display technologies possible, since light from a primary light source must be transformed to other colors in order to achieve variable colors in displays or accurately reproduce white light. We have explored the application of surfactant stabilized organic nano-dots as color conversion filters by fabricating films comprising organic nano-dots dispersed in a carrier polymer of polyvinyl alcohol (PVA). Organic nano-dot dispersions were mixed with PVA solutions and processed to make uniform films. A blue LED with 400 nm emission wavelength was used (Fig. 4a) to excite films containing organic nano-dots. In order to minimize radiation losses, experiments were performed inside an integrating sphere. The color conversion efficiency was calculated by comparing the radiant power of the nano-dot films and pristine PVA films of the same thickness without nano-dots (Supplementary Fig. 21).

To perform a comparative study, we prepared CCLs using the fluorophores described in previous experiments, as well as a high-performance, green TADF emitter, 4CzIPN, which has a PLQY of over 95%. A LED with 400 nm emission wavelength was used as a primary light source simulating the type of light source used in commercial lighting or display applications. The CCE was calculated and compared to a reference color conversion filter of inorganic green fluorescent QDs (InP/ZnSe/ZnS structure, 69.7% PLQY) dispersed in a poly(methyl methacrylate) (PMMA) host, which is representative of the current state-of-the-art in inorganic color conversion filters[50,51]. To make a direct

comparison, the molar concentration of QDs in the CCL film, host polymer concentration, and film casting procedure were kept identical to the nano-dot films.

Compared to the reference QD filter, which converted 21.0% of the incident blue light at 400 nm into green light of 527 nm, 4tBuMB and 4CzIPN showed similar color conversion efficiency, converting it to red or green light with 27.8% or 19.3% efficiency, respectively. Since the intrinsic PLQY of Ttrz-DI (PLQY~ 70%) is less than 4CzIPN, 4CzIPN showed higher color conversion efficiency than Ttrz-DI under similar conditions (Fig. 4b, Supplementary Fig. 21). Among all the organic nano-dots used in this study, the highest CCE of 31% was obtained by TNAP, which has a relatively high PLQY of 89% with a narrow FWHM (35.4 nm).

Blue CCLs are not generally used in color conversion applications, since the incident light is usually blue to begin with. However, in the interest of demonstrating that organic nano-dot CCLs can be used to cover a full gamut of colors, we also fabricated a blue-emitting CzDABNA based CCL, using the same surfactant-assisted precipitation method and calculated its CCE. Although the blue CCL was less efficient compared to the other CCLs (converting ~4% of the 400 nm incident light to 450 nm blue light), it demonstrates that the organic nano-dots can be used to make CCLs even with blue emission. Since boron-containing fluorophores tend to show narrow FWHMs, nano-dots based on materials like CzDABNA show great promise as emissive layers in display applications[52].

In order to quantify the advantages of CCLs based on NDs, we have performed a study comparing the performance and efficiency of nano-dot-based CCLs vs bulk organic fluorophores. Organic fluorophores were dissolved in a PMMA matrix with the same molar concentration as used in the nano-dot CCLs. These results are summarized in Supplementary Table 2. This experiment clearly showed the superiority of nano-dot films over bulk fluorophore CCLs. We have also tested the CCE performance of

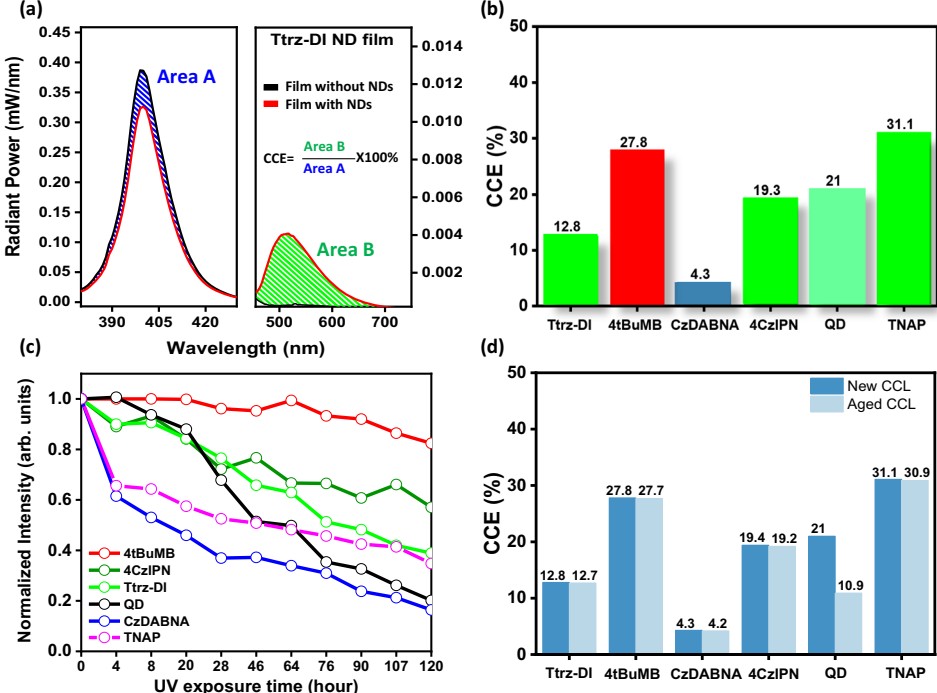

**Fig. 4 Color conversion characteristics. a** Color conversion efficiency (CCE) of Ttrz-DI nano-dot films used as color conversion layers. **b** Color conversion efficiency comparison of 4CzIPN nano-dot film, Ttrz-DI nano-dot film, 4tBuMB nano-dot film, and QD film. The color of each bar corresponds to the emission color of each fluorophore. **c** Stability test of nano-dot films under constant UV exposure. **d** Color conversion performance stability of new and aged films.

bulk fluorophores coated onto a glass substrate without any polymeric host; the CCE of these films was greatly inferior to the ND films.

Supplementary Fig. 22a compares the FWHM values of all the fluorophores used in this study where, CzDABNA showed the narrowest FWHM of 21.1 nm and Ttrz-DI showed the widest FWHM of 87.4 nm along with 4CzIPN (74.4 nm), 4tBuMB (36.8 nm), and TNAP (35.4 nm) having intermediate FWHM values. A color gamut (CIE-1931) was computed using the PL spectra of the nano-dots in Supplementary Fig. 22b, which shows that organic nano-dot synthesized by this method covers a wide gamut in the RGB color space comparable to the present-day commercial NTSC standard color gamut used in displays[53]. Boron-containing nano-dots like TNAP and CzDABNA with narrow FWHM and high color purity (obtained from CIE coordinates) could find merit as potential candidates in display application since their color gamut occupies 105.96% area compared to NTSC color space in the CIE coordinates. Whereas, nano-dots of TADF materials like Ttrz-DI and 4CzIPN with broader FWHM show 71.05% and 76.91% area compared with NTSC color space, would be applicable as CCLs in white LEDs for future lighting purposes. Table 1 summarizes the PLQY, CCE, and related properties for different dispersions and films.

The stability of CCLs is a primary concern, as CCLs must withstand thousands of hours of continuous use in order to make marketable displays or light sources, and many emerging emissive materials, such as inorganic QDs, despite having excellent photophysical properties, cannot be used commercially owing to instability. Therefore, the photo-stability of the nano-dot films was quantified under accelerated aging conditions in which the filters were exposed to constant UV radiation for 120 hours while their CCE with excitation at 400 nm was measured periodically. Compared with the bulk Ttrz-DI film, the nano-dot film of this material showed a much slower decay rate under UV radiation (Supplementary Fig. 23). Figure 4c compares the stability of different CCLs over time. The 4tBuMB nano-dot CCL showed the greatest stability among the materials. Less than 20% decay was observed whereas, 4CzIPN, Ttrz-DI, TNAP, QD and, CzDABNA showed 40%, 60%, 65%, 80%, and 85% decay, respectively, after 120 hours of UV exposure. It is clear from this data that most of the organic nano-dot films showed slower decay characteristics compared to the inorganic QD films.

In another experiment, we investigated the long-term performance stability of the CCL films. In this experiment, freshly fabricated CCLs were stored under ambient conditions for over a month and their CCE was monitored. As shown in Fig. 4d, The CCLs containing the nano-dots show only a very marginal decline of CCE of 1–3%, however, reference CCLs using QDs showed a 52% decrease in CCE after one month of storage. Similar experiments were designed using bulk fluorophore films to compare their data with the nano-dot films. CCE using bulk fluorophore films showed consistently lower efficiency and significant degradation over 1 month (30–60% relative decrease in CCE) compared with the nano-dot films (0.4–2.3% relative decrease) as shown in Supplementary Fig. 24. Thus, CCLs using organic nano-dots exhibited superior stability compared to both inorganic QD films and bulk fluorophore films, confirming their outstanding potential for practical color conversion applications in LEDs, displays, and light sources.

In conclusion, this work shows that surfactants, both ionic and non-ionic, have profound effects on the size, uniformity, and yield of organic nano-dots prepared by the anti-solvent precipitation method. Nano-dots of a variety of fluorophores, including Ttrz-DI, CzDABNA, 4tBuMB, TNAP, and, 4CzIPN were prepared by mixing either ionic or non-ionic surfactants with solutions of the fluorophores in organic solvents prior to precipitation, resulting in uniform, sub-micron particles with close to 100% yield. The effect of the surfactant concentration on the size and morphology of the organic nano-dots was thoroughly characterized and revealed that above the CMC of each surfactant, particle diameter and yield improved dramatically, with 6 mM concentration of either ionic or non-ionic surfactants (~13 or 26 times the CMC, respectively) producing optimal nano-dot dispersions with sizes ranging from 120 to 165 nm. The resulting water-based nano-dot dispersions were easily prepared and could be processed into neat films or dispersed in carrier polymers such as PVA. Such films showed outstanding color conversion characteristics with excellent air and photo-stability as well as high color purity (FWHM from 21.1 to 87.4 nm). The organic nano-dot CCLs showed superior CCE and stability compared with state-of-the-art InP-based QDs with similar PLQY. Among these nano-dots, boron-containing TNAP allowed 105.96% area coverage in the color gamut (CIE-1931) compared with the standard NTSC color space; revealing their potential applicability in modern display devices. Whereas, nano-dots based on TADF materials with broader emission spectra exemplify merits as efficient white LEDs. CCLs made from these nano-dots are also potentially cost-effective compared to their inorganic counterparts due to their superior stability, similar synthetic cost, and reduced environmental/safety concerns associated with water-based processing. By choosing appropriate fluorophores, we show that CCLs based on organic nano-dots prepared by this method have superior characteristics and potential as environmentally friendly replacements for conventional CCL materials, with the potential to greatly impact the futures of display and white light-emitting devices.

## Methods

**Synthesis of tetrabutylammonium oleate (TBA oleate)**. TBA oleate was synthesized by reacting tetrabutylammonium hydroxide with oleic acid. In all, 10 mL of 1.0 M tetrabutylammonium hydroxide in methanol solution was taken in a round bottom flask followed by the addition of 10 mL methanol. Oleic acid was added to the flask dropwise and the reaction was allowed to proceed for 1 hour at room temperature with constant stirring. After the completion of the reaction, the product was separated by washing with diethyl ether to remove unreacted oleic

**Table 1 Characteristics of the fluorescent organic nano-dots.**

| Materials | Average particle size (nm)[a] | PLQY (%)[b] | | | CCE (%)[c] | Yield (%)[d] | |
|---|---|---|---|---|---|---|---|
| | | Film | Dispersion | Solution | | <200 nm | <450 nm |
| Ttrz-DI | 120 | 85.9 | 43.4 | 70 | 12.8 | 96.3 | 98.4 |
| CzDABNA | 110 | 43 | 38.8 | 61.8 | 4.3 | 98.5 | 98.7 |
| 4tBuMB | 162 | 100 | 100 | 100 | 27.8 | 99.6 | 99.9 |
| TNAP | 165 | 84.7 | 60.7 | 89.4 | 31.1 | 98.8 | 99.8 |

[a]Particle sizes measured by optical microscope.
[b]An excitation wavelength of 300 nm was used.
[c]400 nm excitation light was used.
[d]Best nano-dot yields were tabulated.

acid. The product was further washed with ether and subjected to drying under vacuum for 5 hours at room temperature

**Fabrication of organic ND CCLs**. First, 1 g of PVA was dissolved in deionized water to make a 10 weight percentage stock solution. Then, 0.3 g of a 1 mM Ttrz-DI nano-dot dispersion was measured and added into 2.7 g of the 10% PVA solution in water. The mixture was sonicated for 10 mins to make the dispersion uniform. Next, 3 mL of the mixture was drop-cast uniformly on a 2.5 cm × 2.5 cm plastic tray and the solvent was evaporated by heating at 60 °C for 4 hours in a convection oven. After solvent evaporation, the film was taken off using tweezers and the thickness of the film was found to be 160 μm measured using digital Vernier calipers.

**Fabrication of inorganic QD CCLs**. PMMA (average M.W. = 120,000, 0.50 g) was dissolved in 5.0 mL chloroform to make a 10 weight percentage stock solution. To 2.7 g of this solution was added 0.30 g of a 1 mM InP QD dispersion in chloroform. The mixture was sonicated for 10 minutes to achieve uniform dispersion. 3 mL of the mixture was then drop-cast uniformly onto a 2.5 cm × 2.5 cm glass substrate and the solvent was evaporated by heating on a hot plate at 40 °C for 3 hours. The thickness of the film was 120 μm measured using digital Vernier calipers.

**Yield calculation**. A series of solutions of each fluorophore in THF was prepared with a known concentration of each fluorophore and suitable optical density (in the range of 0.05–2 absorbance units) and the absorbance was recorded. These data were used to prepare a calibration curve (absorbance vs concentration) for each fluorophore in THF based on the Beer-Lambert law where concentration and absorbance are linearly proportional. Concentrations of nano-dots in aqueous dispersions (after passing through 0.2 μm and 0.45 μm filters) were determined spectrophotometrically by carefully diluting the aqueous dispersions by a known ratio in a large excess of THF using a micropipette, yielding homogenous THF solutions with optical density suitable for UV-vis. UV-Vis spectra of these dilute solutions were then compared to the calibration curve to back-calculate the molar concentration of each fluorophore in the filtered, aqueous dispersions. The yield calculation was based on the number of moles of the particles which passed through a filter (0.2 μm and 0.45 μm) relative to the total moles of fluorophore initially dissolved in THF prior to anti-solvent precipitation.

## Data availability

All of the data generated in this study have been deposited in Mendeley Data and can be accessed freely through the Mendeley Data website: [https://data.mendeley.com/] Khan, Yeasin (2022), "Data-Synthesis of fluorescent organic nano-dots and their application as efficient color conversion layers", Mendeley Data, V1, doi: 10.17632/zjhzs9wff7.1

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

## Acknowledgements

This work was funded by BK21 FOUR Program (Department of Information Display, Kyung Hee University) funded by the Ministry of Education (MOE, Korea) and the National Research Foundation of Korea (Grant No. NRF-2019M3D1A2104019). This work was also partly supported by the Technology Innovation Program (20006464) funded by the Ministry of Trade, Industry & Energy (MOTIE, Korea)

## Author contributions

Y.K. carried out the nano-dot synthesis and characterizations. R.B. carried out the synthesis of fluorescent materials. S.H. fabricated the color conversion films with technical support provided by Y.J. All the experiments and characterizations were conducted under the supervision of B.W. and J.K.

## Competing interests

The authors declare no competing interests.
