## [Peer Review File · Nature Communications]

Synthesis of fluorescent organic nano-dots and their application as efficient color conversion layersREVIEWER COMMENTS

Reviewer #1 (Remarks to the Author):

The researchers synthesized organic nanodots by the anti-solvent precipitation method with the help of both ionic and non-ionic surfactants. They investigated the surfactants' effect on size, uniformity and yield of nanodots. A series of fluorophores have been used, such as Ttrz-DI, CzDABNA, 4tBuMB, TNAP, and 4CzIPN, and were mixed with the surfactants, and then the uniform sub-micron particles with high yield were obtained. Later on, they found the films with nanodots showed good properties for CEE, regarding, the photostability, color purity and quantum efficiency. The work is solid and shows potential use for color conversion layers. I have several comments on the work, before considering the acceptance of the manuscript.

1. The organic nanodots have some similarity of fluorescent carbon dots, carbonized polymer dots, the researchers should include this topic in the introduction.
2. Several fluorophores were used to prepare the nanodots and films. I am afraid this may cause the high cost of materials as the complicated organic dyes are generally expensive. Regarding the cost, does this new material show potential to replace the currently investigated and used ones?
3. Does the nanodots preparation affect the fluorophore's properties? Or the fluorophores were just doped in the surfactant matrix?
4. If the fluorophore's properties were changed during the nanodots synthesis, the method to measure the yield is unreliable. In addition, the researchers obtained the yield using absorption based approaches, and assumed the efficiency of nanodots passed through a filter. How does the passing efficiency obtain? In addition, to have the yield more reliably, it is necessary to measure the fluorophore concentrations of fluorophore/surfactants preparation mixtures after removal of nanodots.
5. In Figure 4c, the stability test of nanodots films under constant UV exposure was shown, it is clear that 4tBuMB film showed good stability even within 120h. Other nanodots films demonstrated obvious stability concerns. Does this mean the stability is not from the nanodots, or the preparation of film or even the proposed preparation film? The good stability of 4tBuMB film is attributed to the fluorophore properties. If this is true, the proposed strategy to prepare nanodots and films presented little importance.
6. Please revise the following sentence in the introduction section, which is not clear. Light-emitting organic materials have immense commercial importance as florescent and phosphorescent dyes.

Reviewer #2 (Remarks to the Author):

In this manuscript, nano-dots of fluorescent organic material was achieved by using ionic and non-ionic surfactants. In addition, the study on the effect of surfactants on the properties of the fluorescent organic nano-dots was also explored. Moreover, the application of the nano-dots as color conversion layers (CCL) was realized. However, there are a lot of problems in this work as follow. Therefore, recommend the paper to be published in Nature Communication after some major revisions.

1. The advantages of organic nanodots over organic molecules as CCL should be stated in the article.
2. This work mainly explores the influence of surfactants on the formation of organic nano-dots, but it is limited to the influence of concentration of surfactants. The effects of time, temperature and solvent were not explored.
3. The experimental process is not detailed, and many data are not provided, such as the detailed preparation process of organic nanodots, the dialysis time, the specifications of the dialysis bag.
4. Since four organic fluorescent molecules are to be used in this work, all relevant data should be provided, such as synthesis method, fluorescence properties, nuclear magnetic spectrum and so on.
5. Why does the concentration of surfactant have such a big effect on the particle size of the nano-dots? Why do the nano-dots have the best fluorescence properties at smallest particle size? These all need to be explained. In addition, the PLQY should be the standard, not PL intensity.
6. 'When other fluorophores were used (4tBuMB, CzDABNA and TNAP), the smallest particle size was also obtained at 6 mM surfactant concentration (Fig S7- S9)'

According to Figure S7, the particle size of CzDABNA is minimum at 8 mM surfactant concentration.

7. About CCL film:

- 1) The physical pictures of the nano-dots films should be given.
- 2) The data for comparison with films prepared with pure organic molecules should be supplemented.
- 3) 'The CCE was calculated, and compared to a reference color conversion filter comprising an equivalent quantity of inorganic green fluorescent quantum dots (QDs)'

How do you determine the equivalent quantity? Equal mass does not mean equal quantity.

8. In Table 1, the reason for the lowest PLQY of nano-dots needs to be explained.

9. About figures:

- 1) Figure 2c and Figure 2g are the same image.
- 2) The scales of Figure 2c-j and Figure S5-S9 need to be unified.
- 3) Figure S8 is obviously different from the others, and the number of nano-dots is very small.
- 4) Particle size statistical distribution data should be provided.
- 5) What does each picture represent in Figure S11.

Reviewer #3 (Remarks to the Author):

The authors reported the fluorescent organic nano dots prepared from the organic conjugated fluorescent molecules and surfactant. They investigated the effect of the ratio of surfactant (TBA and Triton X-100) on the fluorescence of nano dots. Both ionic and non-ionic surfactant have similar effect. They also put the fluorescent nano dots into the polymer matrix and demonstrated color conversion layer for convert the low wavelength to high wave length. However, the following concerns need to be addressed.

1. The novelty of this manuscript is open to discuss. This strategy has been developed for long period such as phase transfer technique. Because the concentration quenching effect, the report on the organic fluorescent nano dot is not frequently. But the AIE in copolymers have been investigated. Even in the this report, I believe, the authors need to control the amount of the organic fluorescent molecules.

2. The morphology controls strongly depends on the concentration of surfactant, To some concentration of surfactant, the micelles will form specific morphology. That is why 6 mM give the best morphology. Thus, the ratio of the fluorescent molecules to surfactant decide the fluorescent intensity due to the concentration of fluorescent molecules in surfactant. It is common issue, even without prove.

3. The yield of nano dots depends on the properties and CMC of surfactant. fluorescent molecules has minor effect.

4. I do not understand why the author define color conversion efficiency (CCE). What is the different between the PLQY and CCE?

5. The author claimed the triton X-100 film in Figure 2e and 2i, it is hard to say film. it may just a layer of micellar particles stack on the substrate.

6. color purity can be obtained from the CIE coordinates. it is not from FWHM

Response to reviewer's comments

Manuscript ID: NCOMMS-21-36390-T

Title: Synthesis of fluorescent organic nano-dots and their application as efficient color conversion layers

We are grateful to the reviewers for their positive comments and useful suggestions. We have revised our present manuscript as per the reviewer's suggestions/comments. The detailed response to reviewer's comments are as follows. The original comments from the referee are in black. The response to the comments is in blue text and the highlighted portion were added in the revised manuscript.

REVIEWER COMMENTS

Reviewer #1 (Remarks to the Author):

The researchers synthesized organic nanodots by the anti-solvent precipitation method with the help of both ionic and non-ionic surfactants. They investigated the surfactants' effect on size, uniformity and yield of nanodots. A series of fluorophores have been used, such as Ttrz-DI, CzDABNA, 4tBuMB, TNAP, and 4CzIPN, and were mixed with the surfactants, and then the uniform sub-micron particles with high yield were obtained. Later on, they found the films with nanodots showed good properties for CEE, regarding, the photostability, color purity and quantum efficiency. The work is solid and shows potential use for color conversion layers. I have several comments on the work, before considering the acceptance of the manuscript.

1. The organic nanodots have some similarity of fluorescent carbon dots, carbonized polymer dots, the researchers should include this topic in the introduction.

[Our response] We thank the reviewer for the suggestion. We've included fluorescent carbon dots and carbonized polymer dots to the introduction on page 3 of the revised manuscript and included 5 new references (new references 36-40):

"In contrast, fluorescent carbon dots and carbonized polymer dots have shown practical importance in white light emitting diodes.³⁶⁻⁴⁰"

36. Wang, Q. *et al.* S,N-Codoped oil-soluble fluorescent carbon dots for a high color-rendering WLED. *J. Mater. Chem. C* **8**, 4343–4349 (2020).
37. Zhang, Y. *et al.* Solid-State Fluorescent Carbon Dots with Aggregation-Induced Yellow Emission for White Light-Emitting Diodes with High Luminous Efficiencies. *ACS Appl. Mater. Interfaces* **11**, 24395–24403 (2019).
38. Wang, Y. *et al.* High color rendering index trichromatic white and red LEDs prepared from silane-functionalized carbon dots. *J. Mater. Chem. C* **5**, 9629–9637 (2017).
39. Wang, Y. *et al.* Polysiloxane Functionalized Carbon Dots and Their Cross-Linked Flexible Silicone Rubbers for Color Conversion and Encapsulation of White LEDs. *ACS Appl. Mater.*

Interfaces **8**, 9961–9968 (2016).

40. Shao, J. *et al.* Full-Color Emission Polymer Carbon Dots with Quench-Resistant Solid-State Fluorescence. *Adv. Sci.* **4**, 1700395 (2017).

2. Several fluorophores were used to prepare the nanodots and films. I am afraid this may cause the high cost of materials as the complicated organic dyes are generally expensive. Regarding the cost, does this new material show potential to replace the currently investigated and used ones?

[Our response] We thank the reviewer for bringing up the important issue of material cost. Currently used color conversion layers use materials with similar costs. For example, inorganic quantum dot CCLs require expensive and often toxic starting materials such as phosphines, cadmium and indium salts, whereas color conversion layers prepared from organic nano-dots would show better efficiency, stability and broader applicability and reduced environmental / safety concerns due to their water based processing conditions. We believe the new approach would have similar or low cost compared to other color converters with higher efficiency and better long-term stability.

We've added the following statement on page 13 of the revised manuscript to clarify this point:

“CCLs made from these nano-dots are also potentially cost effective compared to their inorganic counterparts due to their superior stability, similar synthetic cost and reduced environmental / safety concerns associated with water-based processing.”

3. Does the nanodots preparation affect the fluorophore's properties? Or the fluorophores were just doped in the surfactant matrix?

[Our response] We thank the reviewer for this comment and we have tried our best to explain the reviewers' queries.

When the surfactant molecules encounter with the fluorescent organic molecules, the hydrophobic tails of the surfactant molecules interacts with the organic molecules due to London dispersion force; whereas the hydrophilic heads dangle at the perimeter of the surrounded organic molecule. Right after the anti-solvent water is added to the mixture, the surfactant forms micelles due to the interaction of the hydrophilic head with polar water molecules resulting encapsulated organic nanoparticles. Formation of the micelles depends directly on the concentration of the surfactant molecules. Therefore, we observe small sized fluorophore particles when the concentration of the surfactant is over the critical micelle concentration of the surfactant. Since there are no chemical reactions involved in this process, the properties of the fluorophores remain almost unaffected.

We've added the following statement on page 3 of the revised manuscript to clarify this point:

“Nano-dots are formed when a solution of fluorophore rapidly precipitates as it is mixed with a non-solvent (water). If this happens in the presence of a surfactant above its CMC, micelles are able to encapsulate the fluorophore particles as they precipitate, through interaction between the non-polar surfactant tails and non-polar fluorophore molecules. The presence of surfactant molecules is expected to minimize the surface energy of small particles, whereas in the absence of surfactant, the surface energy is minimized when large fluorophore particles are formed; thus small particles (nano-dots) are formed when the precipitation occurs in the presence of a surfactant above its CMC. Since the formation of nano-dots involves a physical change in the state of the fluorophore (no chemical transformation occurs), most of the properties of the fluorophore are not significantly changed, though the absorption and emission spectra of such nano-dot dispersions fall between those of the bulk, solid state phase and

true solution phase of the fluorophore, as can be seen in the photoluminescence (PL) spectra of nano-dots.”

4. If the fluorophore’s properties were changed during the nanodots synthesis, the method to measure the yield is unreliable. In addition, the researchers obtained the yield using absorption based approaches, and assumed the efficiency of nanodots passed through a filter. How does the passing efficiency obtain? In addition, to have the yield more reliably, it is necessary to measure the fluorophore concentrations of fluorophore/surfactants preparation mixtures after removal of nanodots.

[Our response] We thank the reviewer for his concern about the yield calculation of the nano-dots. In the above discussion, we clarified that the fluorophore’s properties and chemical structure were unaffected apart from their reduction in size. Therefore, the yield calculation was possible by an spectrophotometric approach. In this approach, we first prepared a series of solutions in THF with a known concentration of each fluorophore and suitable optical density (0.05 to 2 absorbance units) and the absorbance was recorded. This data was used to prepare a calibration curve (absorbance vs concentration) at various concentrations for each fluorophore in THF based on Beer-Lambert’s law where concentration and absorbance are linearly proportional to each other. Based on that calibration curve it was possible to measure the concentration of fluorophore present in the nano-dot dispersions as follows. Nano-dots of each material were prepared, subjected to filtration through a 450 nm filter or a 200 nm filters, then diluted by a known proportion using a large excess of THF to re-dissolve the fluorophore. The corresponding filters would retain any particles having larger size than 450 nm or 200 nm, respectively. Absorbance spectra of the nano-dots that passed through the filters and were diluted with THF were recorded and compared to the previously described calibration curve for each fluorophore. The yield was calculated based on the moles of nano-dots passing through the filters to that of the moles of the original fluorophore.

This procedure has been described in detail in the experimental section as follows:

“A series of solutions of each fluorophore in THF were prepared with a known concentration of each fluorophore and suitable optical density (in the range of 0.05 to 2 absorbance units) and the absorbance was recorded. This data was used to prepare a calibration curve (absorbance vs concentration) for each fluorophore in THF based on the Beer-Lambert law where concentration and absorbance are linearly proportional. Concentrations of nano-dots in aqueous dispersions (after passing through 0.2 μm and 0.45 μm filters) were determined spectrophotometrically by carefully diluting the aqueous dispersions by a known ratio in a large excess of THF using a micropipette, yielding homogenous THF solutions with optical density suitable for UV-vis. UV-Vis spectra of these dilute solutions were then compared to the calibration curve to back-calculate the molar concentration of each fluorophore in the filtered, aqueous dispersions. The yield calculation was based on the number of moles of the particles which passed through a filter (0.2 μm and 0.45 μm) relative to the total moles of fluorophore initially dissolved in THF prior to anti-solvent precipitation.”

5. In Figure 4c, the stability test of nanodots films under constant UV exposure was shown, it is clear that 4tBuMB film showed good stability even within 120h. Other nanodots films demonstrated obvious stability concerns. Does this mean the stability is not from the nanodots, or the preparation of film or even the proposed preparation film? The good stability of 4tBuMB film is attributed to the fluorophore properties. If this is true, the proposed strategy to prepare nanodots and films presented little importance.

[Our response] We thank the reviewer for the query about the stability of the nano-dot films. The property of the fluorophore definitely has an impact on the stability of the films. However, the nano-dot films showed improved stability when compared with the bulk fluorophore films of the same material as reported in the supporting information figure S21.

Figure S21. Stability test comparison of Ttrz-DI ND films vs bulk Ttrz-DI film under constant UV exposure.”

We have also checked the color conversion efficiency of aged (over a month) films prepared by nano-dots and bulk organic fluorophore films and found that the color conversion efficiency of the bulk films was reduced significantly, on the other hand only a slight reduction was observed in the nano-dot films as quoted in the following (page 13 of the manuscript):

“Similar experiments were designed using bulk fluorophore films to compare their data with the nano-dot films. CCE using bulk fluorophore films showed consistently lower efficiency and significant degradation over one month (30-60% relative decrease in CCE) compared to the nano-dot films (0.4-2.3% relative decrease) as shown in figure S22. Thus, CCLs using organic nano-dots exhibited superior stability compared to both inorganic QD films and bulk fluorophore films.”

Figure S22. Comparison of color conversion efficiency for newly fabricated CCLs and CCLs aged for one month based on nano-dot (ND) films and bulk fluorophore films.

6. Please revise the following sentence in the introduction section, which is not clear. Light-emitting organic materials have immense commercial importance as florescent and phosphorescent dyes.

[Our response] We thank the reviewer for suggesting to revise this ambiguous sentence. We have

revised the sentence as follows:

“Fluorescent and phosphorescent organic dyes have immense commercial importance in light emission applications”.

Reviewer #2 (Remarks to the Author):

In this manuscript, nano-dots of fluorescent organic material was achieved by using ionic and non-ionic surfactants. In addition, the study on the effect of surfactants on the properties of the fluorescent organic nano-dots was also explored. Moreover, the application of the nano-dots as color conversion layers (CCL) was realized. However, there are a lot of problems in this work as follow. Therefore, recommend the paper to be published in Nature Communication after some major revisions.

1. The advantages of organic nanodots over organic molecules as CCL should be stated in the article.

[Our response] We thank the reviewer for the valuable suggestion. In order to show the advantage of using nano-dots over bulk organic molecules we have designed an experiment where we prepared color conversion films with bulk fluorophores in polymethylmethacrylate (PMMA) matrix, maintaining the same concentration as the nano-dot CCLs, as well as pure fluorophore films, without additives, on glass substrates. The color conversion of the nano-dots showed superior performance than the other approaches. We have added the following text describing these experiments on page 12 of the revised manuscript as well as included relevant details in the experimental section:

“In order to demonstrate the advantages of CCLs based on NDs, we’ve performed a study quantifying the performance of nano-dot based CCLs compared to bulk organic fluorophores. Organic fluorophores were dissolved in a PMMA matrix with the same molar concentration as used in the nano-dot CCLs. These results are summarized in **Table S2**. This experiment clearly showed the superiority of nano-dot films over bulk fluorophore CCLs. We have also tested the CCE performance of bulk fluorophores coated on a glass substrate without any polymeric host; the CCE of these films was greatly inferior to the ND films.”

Table S2. Comparison between ND CCLs, bulk fluorophore in PMMA CCLs and bulk fluorophore CCLs without additives.

Fluorophores	CCE (%)		
	ND films	Bulk fluorophores in PMMA	Bulk fluorophores without matrix
Ttrz-DI (Green)	12.80	7.25	1.46
4tBuMB (Red)	27.80	14.03	1.35
CzDABNA (Blue)	4.30	1.19	0.33
TNAP (Green)	31.10	14.8	2.45
4CzIPN (Green)	19.30	9.19	1.77

2. This work mainly explores the influence of surfactants on the formation of organic nano-dots, but it is limited to the influence of concentration of surfactants. The effects of time, temperature and solvent were not explored.

[Our response] We thank the reviewer for the comment and we have investigated the effect of solvents, anti-solvents and temperature as suggested by the reviewer. We have discussed these experimental findings in the main text as well as updated the supporting information with figures S15-S18 as follows:

On page 9 of the main text:

“To develop a more complete picture of how different conditions affect nano-dot properties, we explored the effect of using different solvents on the yield of Ttrz-DI nano-dots. The concentration of Ttrz-DI and Triton X-100 was fixed at 0.5 mM and 6 mM respectively with water as the anti-solvent. Apart from THF, other water-miscible solvents including 1,3-dioxalane, 1,4-dioxane, N-methylpyrrolidone (NMP) and N-methyl imidazole (NMI) were used for the nano-dot synthesis. A slight increase in yield and PLQY was obtained in the case of 1,3-dioxalane and NMP however, the average particle size was found to be the lowest when the solvent medium was THF (figure S16). Different polar anti-solvents like methanol, ethanol, iso-propanol, and 2-methoxyethanol were also investigated for the ND synthesis. In this case, the concentration of Ttrz-DI and Triton X-100 was kept constant in THF solvent medium. Large-sized visible particles with negligible nano-dot yields were obtained when these anti-solvents were used. Water showed the best yield among all of the investigated anti-solvents (Figure S17). The injection temperatures of the anti-solvent were another important parameter for investigation. Temperature of the anti-solvent (water) ranging from 25 °C (room temperature) to 90 °C was taken into consideration for this experiment. Yield and PLQY of the nano-dots were found to decrease at increasing anti-solvent injection temperatures and the best yield was obtained at the room temperature (Figure S18). Therefore, optimal conditions for nano-dot synthesis included using THF as a solvent and water as the anti-solvent at room temperature.”

New figures in the supporting information:

“

Figure S16. Effect of different solvents on the synthesis of Ttrz-DI nano-dots. Optical microscope images and particle size distribution graphs of Ttrz-DI NDs prepared using (a, b) 1,3-dioxalane, (c, d) 1,4-dioxane, (e, f) N-methylimidazole (NMI), (g, h) N-methylpyrrolidone (NMP), (i, j) Tetrahydrofuran (THF) as solvents; (k) Table showing their mean, standard deviation, minimum and maximum sizes; (l) PLQY of the Ttrz-DI nano-dots prepared using the corresponding solvents; (m) nano-dot yields.

Figure S17. Effect of anti-solvents on the synthesis of Ttrz-DI nano-dots. Optical microscope images and particle size distribution graphs of Ttrz-DI NDs prepared using (a, b) methanol, (c, d) iso-propanol (IPA), (e, f) Ethanol, (g, h) 2-methoxyethanol (2-MeEtOH), and (i, j) water as an anti-solvent; (k) Table showing their mean, standard deviation, minimum and maximum sizes; (l) PLQY of the Ttrz-DI nano-dots prepared using the corresponding anti-solvents; (m) picture showing large particle size of Ttrz-DI when IPA was used as anti-solvent; (n) picture showing large particle size of Ttrz-DI when methanol was used as anti-solvent; (o) nano-dot yields.

Figure S18. Effect of anti-solvent (water) injection temperature on the synthesis of Ttrz-DI nano-dots. Optical microscope image and particle size distribution graphs of Ttrz-DI NDs prepared by injecting anti-solvent water at (a, b) room temperature, (c, d) 50 °C, (e, f) 60 °C, (g, h) 70 °C, (i, j) 80 °C, and (k, l) 90 °C; (m) Table showing their mean, standard deviation, minimum and maximum sizes; (n) nano-dot

yields; (o) PLQY of the Ttrz-DI nano-dots prepared by injecting into anti-solvent (water) at variable temperatures.”

The effect of time (aging) has been included in the supporting information section as figure S15.

Figure S15. Ttrz-DI ND aggregation vs time comparison using either Triton X-100 or TBAOleate surfactants at 6 mM concentration.

Including the following description:

“To characterize how Ttrz-DI ND dispersions were affected by aging, dispersions were prepared for both ionic and nonionic surfactants where the concentration of Ttrz-DI was kept at 0.01 mM. ND films were prepared by drop casting the freshly made dispersions over glass substrates. The dispersions were allowed to age for one hour and drop casted on additional glass substrates. The process was then repeated for 2 to 12 hours and the particle sizes were measured by optical microscopy. The average particle size increased from 120 nm to 1.9 μm for Triton X-100 and 144 nm to 0.8 μm for TBAOleate within 12 hours period. From the data, the aggregation rate of Ttrz-DI NDs was found greater in the case of Triton X-100 than that of TBAOleate surfactant.”

3. The experimental process is not detailed, and many data are not provided, such as the detailed preparation process of organic nanodots, the dialysis time, the specifications of the dialysis bag.

[Our response] We thank the reviewer for pointing out this shortcoming in the manuscript. In response, we’ve added a number of additional details in the supporting information as well as in the main text to ensure that the procedure is as clear and easily repeatable as possible. Changes have been highlighted in yellow to make these changes clear.

The following additional information has been added to the Experimental section of the supporting information:

“Cellulose acetate dialysis tubing with a molecular weight cut-off 14,000 Dalton was purchased from Sigma-Aldrich.”

The following changes have been made to the Nano-dot synthesis and characterization section of the

main text:

“Then, 0.1 mL of the fluorophore solution was taken in a vial followed by the addition of a variable amounts (0.01, 0.04, 0.1, 0.2, 0.3, 0.4, 0.5 mL) of the surfactant solution. Excess THF was added to keep the total volume constant at 0.6 mL. After that, 4.4 mL of deionized water was injected rapidly into the fluorophore-surfactant solution to make dispersions of nano-dots. The total volume was maintained constant at 5 mL after mixing with non-solvent. The dispersions were filtered through Polytetrafluoroethylene (PTFE) syringe filters with different pore sizes (450 nm and 200 nm diameter) and subjected to dialysis using cellulose acetate tubes for 12 hours to remove the excess surfactant from the dispersions. The dispersions were then concentrated under a reduced pressure of around 0.01 Torr, using a Schlenk line, to evaporate 90% of the solvents.”

4. Since four organic fluorescent molecules are to be used in this work, all relevant data should be provided, such as synthesis method, fluorescence properties, nuclear magnetic spectrum and so on.

[Our response] We thank the reviewer for the concern about including all relevant material data. The synthesis and characterization data of Ttrz-DI and TNAP have already been included in the supporting information section. The synthesis and properties of the other organic fluorophores (CzDABNA, 4tBuMB and 4CzIPN) have already published elsewhere and we have cited these articles.

The references related to CzDABNA, 4tBuMB and 4CzIPN fluorophores have been included as references 45-47 of the revised manuscript:

CzDABNA- Oda, S. et al. Carbazole-Based DABNA Analogues as Highly Efficient Thermally Activated Delayed Fluorescence Materials for Narrowband Organic Light-Emitting Diodes. *Angew. Chemie Int. Ed.* 60, 2882–2886 (2021).

4tBuMB- Jung, Y. H. et al. A New BODIPY Material for Pure Color and Long Lifetime Red Hyperfluorescence Organic Light-Emitting Diode. *ACS Appl. Mater. Interfaces* 13, 17882–17891 (2021).

4CzIPN- Uoyama, H., Goushi, K., Shizu, K., Nomura, H. & Adachi, C. Highly efficient organic light-emitting diodes from delayed fluorescence. *Nature* 492, 234–238 (2012).

The synthesis and characterization part for Ttrz-DI and TNAP have been included in the supporting information as follows:

“II. Synthesis of 5,10,15-tris(4-(4,6-diphenyl-1,3,5-triazin-2-yl)phenyl)-10,15-dihydro-5H-diindolo[3,2-a:3',2'-c]carbazole (Ttrz-DI)

“2-(4-bromophenyl)-4,6-diphenyl-1,3,5-triazine (Trz, 8.85 mmol) and 10,15-dihydro-5H-diindolo[3,2-a:3',2'-c]carbazole (fused carbazole, 2.95 mmol) was mixed in a 3:1 molar ratio in the presence of tris(dibenzylideneacetone)dipalladium(0) (Pd₂(dba)₃, 0.08mmol), tri-*tert*-butylphosphine (*t*Bu₃P, 1.86 mmol), sodium *tert*-butoxide (NaO*t*Bu, 5.15mmol) and anhydrous *o*-xylene. All the reagents were taken into a two neck round bottom flask equipped with a condenser. The mixture was flushed with nitrogen and subjected to vacuum several times to create an inert atmosphere. Then, the reaction was heated with constant stirring at 135 °C for 12 hours under reflux. After reaction completion, the mixture was extracted with dichloromethane and deionized water several times. The aqueous phase was discarded and the dichloromethane layer was dried over anhydrous sodium sulphate. The mixture was then filtered and the remaining solvents were removed by rotary evaporation. The final product, 5,10,15-tris(4-(4,6-diphenyl-1,3,5-triazin-2-yl)phenyl)-10,15-dihydro-5H-diindolo[3,2-a:3',2'-c]carbazole (Ttrz-DI) was isolated by silica gel column chromatography. The final product was analyzed by ¹H NMR spectroscopy in CDCl₃ (Fig. S1).”

Scheme S1. Reaction scheme for the synthesis of *Ttrz-DI*

Figure S1. ^1H NMR data of the synthesized *Ttrz-DI*.

Characterization of *Ttrz-DI*

We first measured the basic photo physical properties of the *Ttrz-DI* chromophore. It showed strong absorption at around 300 nm wavelength region which corresponds to the $\pi\rightarrow\pi^*$ transition of carbazole and phenyl groups with $n\rightarrow\pi^*$ transition around 382 nm. The bandgap of the material was calculated to be 2.82 eV from the absorption onset (440 nm). Photoluminescence spectra obtained in THF solvent show a maximum at 544 nm along with solvatochromism (Fig. S2a) due to strong charge transfer characteristics.

Figure S2. (a) UV-Vis, RTPL and LTPL spectra of *Ttrz-DI* in toluene, (b) Solvatochromism of *Ttrz-DI* in different solvents.

The maximum photoluminescence (PL) peak of *Ttrz-DI* in hexane was observed at 475 nm and the peak position was red-shifted when PL was measured in toluene, acetonitrile, tetrahydrofuran, and methanol, respectively. The onset value from the room temperature PL (RTPL) revealed that the singlet energy of

Ttrz-DI is around 2.11 eV while the triplet energy of 2.34 eV was calculated from low-temperature PL (LTPL) analysis in toluene solvent at 77K. The energy gap between singlet and triplet state (0.23 eV) was obtained by subtracting the triplet energy from singlet energy of Ttrz-DI (Fig. S2b).

Figure S3. Cyclic voltammogram of Ttrz-DI

The highest occupied molecular orbital (HOMO) of Ttrz-DI was found to be -5.15 eV, calculated from its cyclic voltammogram (Fig. S3). Additionally, the respective lowest unoccupied molecular orbital (LUMO) of -2.33 eV was obtained by adding the bandgap value from UV-Vis absorption data.

III. Synthesis of *N,N,6,10-tetra(naphthalen-2-yl)-6,10-dihydro-6,10-diaza-16b-boraanthra[3,2,1-de]tetracen-8-amine (TNAP)*

1,3,5-tribromobenzene (TBB, 1 g, 3.18 mmol), di(naphthalen-2-yl)amine (NAPA, 2.74 g, 10.17 mmol), tris(dibenzylideneacetone)dipalladium(0) Pd₂(dba)₃ (0.26 g, 0.28 mmol), tri-*tert*-butylphosphine (P(*t*-Bu)₃, 0.8 mL, 3.17 mmol) and sodium *tert*-butoxide (NaO*t*Bu, 1.84 g, 19.15 mmol) were added in a two neck round bottom flask equipped with condenser. Then, anhydrous toluene (60 mL) was added and the mixture was refluxed at 106°C for 18 hours. After completion of the reaction, mixture was filtered and washed with hexane, then recrystallized from dichloromethane and *n*-Hexane to obtain the intermediate of *N,N',N'',N''',N''',N''''*-hexa(naphthalen-2-yl)benzene-1,3,5-triamine (TNA) an yield of 79.35%.

Afterwards, TNA (500 mg, 0.57 mmol) was dissolved in 10 mL of *o*-dichlorobenzene at room temperature in a two neck flask equipped with reflux condenser and magnetic stirring bar. Boron tribromide BBr₃ (0.18 g, 0.74 mmol) was added drop-wise under inert conditions. Then, the reaction mixture was stirred at 180 °C for 24 hours, and the reaction progress was monitored by thin layer chromatography. After completion, reaction mixture was extracted with water and chloroform three times (150 mL), and the organic layer was dried over anhydrous sodium sulphate. After concentration under reduced pressure, the crude mixture was purified by silica gel column chromatography using *n*-hexane: dichloromethane (7:1) as a mobile phase and the product (TNAP) was collected in 23.8 % yield.

Scheme S2. Reaction scheme for the synthesis of TNAP

Characterization of TNAP

Orange color solid; ¹H NMR (400 MHz, CD₂Cl₂-d₂) δ (ppm) 9.75 (s, 2H), 8.23-8.26 (m, 2H), 7.79-7.85 (m, 4H), 7.69 (d, J=8.8 Hz, 4H), 7.52-7.55 (m, 4H), 7.42-7.49 (m, 10H), 7.34-7.41 (m, 8H), 7.30 (d,

$J=8.8$ Hz, 2H), 7.19 (s, 2H), 7.05 (dd, $J=8.0, 2.4$ Hz, 2H), 5.79 (s, 2H); ^{13}C NMR (100 MHz, $\text{CD}_2\text{Cl}_2\text{-d}_2$) δ (ppm) 152.4, 149.2, 146.0, 143.9, 139.8, 136.4, 135.4, 134.8, 134.1, 133.0, 131.3, 130.7, 129.4, 129.1, 128.8, 128.6, 128.1, 128.0, 127.8, 127.5, 127.2, 127.1, 126.9, 126.7, 126.5, 125.3, 125.2, 123.7, 122.9, 112.5, 99.0.

Figure S4. (a) UV-Vis absorption and PL emission data, (b) RTPL and LTPL spectra, (c) Solvatochromism in different solvents, (d) Cyclic voltammogram of TNAP-DABNA

The optical bandgap of TNAP was calculated to be 2.375 eV from the absorption onset (522 nm). Photoluminescence spectra obtained in THF solution showed a peak at 523 nm along with solvatochromism (Fig. S4c) due to strong charge transfer characteristics. The maximum photoluminescence (PL) peak of Ttrz-DI in hexane was obtained at 510 nm and the peak position was red-shifted when PL was measured in toluene, tetrahydrofuran, and methylene chloride respectively. The onset value from the room temperature PL revealed that the singlet energy of TNAP-DABNA is around 2.54 eV while a triplet energy of 2.28 eV was calculated from low-temperature PL (LTPL) analysis in toluene solvent at 77 K. The energy gap between singlet and triplet states (0.26 eV) was obtained by subtracting the triplet energy from singlet energy of Ttrz-DI (Fig. S4b). The HOMO of Ttrz-DI was found to be -5.43 eV calculated from its cyclic voltammogram (Fig. S4d). Additionally, the respective LUMO of -3.05 eV was obtained by adding the bandgap value from UV-Vis absorption data. ”

5. Why does the concentration of surfactant have such a big effect on the particle size of the nano-dots? Why do the nano-dots have the best fluorescence properties at smallest particle size? These all need to be explained. In addition, the PLQY should be the standard, not PL intensity.

[Our response]

When the surfactant molecules encounter with the fluorescent organic molecules, the hydrophobic tails of the surfactant molecules interacts with the organic molecules due to London dispersion force; whereas the hydrophilic heads dangle at the perimeter of the surrounded organic molecule. Right after the anti-solvent water is added to the mixture, the surfactant forms micelles due to the interaction of the hydrophilic head with polar water molecules resulting encapsulated organic nanoparticles. This micelle formation plays a critical role in determining the size of the nanoparticle which largely depends on the concentration of the surfactant molecules. Therefore, we observe small sized particles when the concentration is above the critical micelle concentration (CMC) and large particles when the concentration remains under the CMC. Also due to this reason we observe increased fluorescence when the concentration of surfactant is above CMC where nano sized fluorophore encapsulated in

micelles floats around in the water dispersion; whereas, we observed reduced or no fluorescence below CMC where micelles cannot form resulting in a precipitation of the bulk organic fluorophores.

We've added the following statement on page 3 of the revised manuscript to clarify this point:

“Nano-dots are formed when a solution of fluorophore precipitates as it is mixed with a non-solvent (water). If this happens in the presence of a surfactant above its CMC, micelles may encapsulate the fluorophore particles as they precipitate, through interaction between the non-polar surfactant tails and non-polar fluorophore molecules. The presence of surfactant molecules is expected to minimize the surface energy of small particles, whereas in the absence of surfactant, the surface energy is minimized when large fluorophore particles are formed; thus small particles (nano-dots) are formed when the precipitation occurs in the presence of a surfactant above its CMC. Since the formation of nano-dots involves a physical change in the state of the fluorophore (no chemical transformation occurs), most of the properties of the fluorophore are not significantly changed, though the absorption and emission spectra of such nano-dot dispersions fall between those of the bulk, solid state phase and true solution phase of the fluorophore, as can be seen in the photoluminescence (PL) spectra of nano-dots.”

6. ‘When other fluorophores were used (4tBuMB, CzDABNA and TNAP), the smallest particle size was also obtained at 6 mM surfactant concentration (Fig S7- S9)’

According to Figure S7, the particle size of CzDABNA is minimum at 8 mM surfactant concentration.

[Our response] We thank the reviewer for pointing out this issue. The reviewer is correct 8 mM surfactant concentration showed the smallest average particle sizes in the case of CzDABNA. We have corrected this statement in the main text. Changes have been highlighted in yellow on page 8 of the revised manuscript and copied below:

“the smallest particle size was also obtained at 6 mM surfactant concentration (Fig S8- S9), with the exception of CzDABNA, which showed the smallest average size of 110 nm at 8 mM surfactant concentration. It should be noted that this was only 3 nm smaller than the size (113 nm) obtained using 6 mM surfactant concentration (Fig S7).”

7. About CCL film:

- 1) The physical pictures of the nano-dots films should be given.
- 2) The data for comparison with films prepared with pure organic molecules should be supplemented.
- 3) ‘The CCE was calculated, and compared to a reference color conversion filter comprising an equivalent quantity of inorganic green fluorescent quantum dots (QDs)’
How do you determine the equivalent quantity? Equal mass does not mean equal quantity.

[Our response] We thank the reviewer for the suggestions to include additional data and information about CCL films, which we agree strengthens the manuscript. We have added this information as suggested by the reviewer as figures S19 and Table S2, and included additional discussion about these data on page 11,

- 1) The physical pictures of the nano-dot films are added to figure S19 in the supporting information

“

Figure S19. Color conversion efficiency of (a) *Ttrz-DI* ND film, (b) *4tBuMB* ND film, (c) *CzDABNA* ND film, (d) *TNAP* ND film, (e) *4CzIPN* ND film, (f) *Inorganic QD* film. Pictures of the nano-dot films are provided in the inset. These films converted radiation with ~400 nm wavelength to their corresponding emission spectra“

2) We have included the comparison data of nano-dot CCLs and bulk fluorophores in the main text (page 12) and in the supporting information quoted as following:

Main text part (page 12)-

“In order to quantify the advantages of CCLs based on NDs, we’ve performed a study comparing the performance and efficiency of nanodot based CCLs vs bulk organic fluorophores. Organic fluorophores were dissolved in a PMMA matrix with the same molar concentration as used in the nanodot CCLs. These results are summarized in Table S2. This experiment clearly showed the superiority of nano-dot films over bulk fluorophore CCLs. We have also tested the CCE performance of bulk fluorophores coated onto glass substrates without any polymeric host; the CCE of these films was greatly inferior to the ND films.”

Supporting information part-

“**Table S2.** Comparison between ND CCLs, bulk fluorophore in PMMA CCLs and bulk fluorophore CCLs without additives.

Fluorophores	CCE (%)		
	ND films	Bulk fluorophores in PMMA	Bulk fluorophores without matrix
Ttrz-DI (Green)	12.80	7.25	1.46
4tBuMB (Red)	27.80	14.03	1.35
CzDABNA (Blue)	4.30	1.19	0.33
TNAP (Green)	31.10	14.8	2.45

3) Equal concentrations (1 mM) of organic nano-dots and inorganic quantum dots were used for this experiment to make a 10% solution of their corresponding polymeric matrices.

We've added the following statement on page 10 of the revised manuscript to address this issue:

"To make a direct comparison, the molar concentration of QDs in the CCL film, host polymer concentration and film casting procedure were kept identical to the nano-dot films."

8. In Table 1, the reason for the lowest PLQY of nano-dots needs to be explained.

[Our response] We thank the reviewer for pointing out that the reason for low PLQY of the nano-dot dispersions was not explained adequately. The following explanation has been provided in the supporting information section under Figure S11.

"the PLQY values of the dispersions were lower in the case of Ttrz-DI, TNAP, and CzDABNA compared to their solutions in THF. The lower PLQY of these materials in aqueous dispersions can be attributed to their TADF emission mechanism. PLQYs of these materials rely on delayed fluorescence involving reverse intersystem crossing, rather than prompt fluorescence (singlet excited to ground state) and it is generally observed that the quantum efficiency of delayed fluorescence decreases especially in polar solvents.³ In our study, the dispersion medium of these TADF materials was water. The electronic interactions of TADF materials with polar water molecules therefore decreased fluorescence quantum efficiency compared to solutions in organic solvents. However, in the absence of polar water molecules, solid films of the organic NDs showed higher quantum yields than their dispersions. In the case of Ttrz-DI the PLQY of the film was around 86% where its dispersion was 43%. Also, in the case of CzDABNA, the PLQY increased slightly from 39% in dispersion to 43% in the film state. TNAP also showed similar behavior where PLQY of dispersion was 60.7% whereas, PLQY in the film state showed 84.7% quantum efficiency. This phenomenon was not observed for 4tBuMB since it is a prompt fluorescent dye compound and is not as strongly affected by interaction with polar solvent molecules; here, both the dispersion and film PLQY were close to 100%. Table 1 in the main text summarizes the PLQY and related properties for different dispersions and films.

Furthermore, we have also cited the following article about the effect of solvents in TADF materials in the supporting information:

Ishimatsu, R. et al. Solvent Effect on Thermally Activated Delayed Fluorescence by 1,2,3,5-Tetrakis(carbazol-9-yl)-4,6-dicyanobenzene. *J. Phys. Chem. A* **117**, 5607–5612 (2013).

9. About figures:

- 1) Figure 2c and Figure 2g are the same image.
- 2) The scales of Figure 2c-j and Figure S5-S9 need to be unified.
- 3) Figure S8 is obviously different from the others, and the number of nano-dots is very small.
- 4) Particle size statistical distribution data should be provided.
- 5) What does each picture represent in Figure S11 (now S12).

[Our response] We thank the reviewer for pointing out these issues and have corrected the concerns as follows.

1) Figure 2c and 2g are the images of Ttrz-DI casted on a glass substrate without any surfactant which are basically same images, which was the same control experiment used for both surfactants. However, in response to the reviewer's suggestion, we have removed 2g and modified the figure 2 in the

manuscript as shown below:

“

Figure 2. Photoluminescence spectra of solution, film and nano-dot dispersion of Ttrz-DI with (a) Triton X-100, (b) TBA oleate. Optical microscope (OM) images of a Ttrz-DI dispersion deposited on a glass substrate synthesized (c) without surfactant and (d) with 6 mM Triton X-100 surfactant. (e) SEM image of a Ttrz-DI ND/Triton X-100 dispersion deposited on a glass substrate. (f) Fluorescence microscope (FM) image of a Ttrz-DI ND/Triton X-100 dispersion deposited on a glass substrate. Optical microscope images of a Ttrz-DI dispersion deposited on glass a substrate synthesized (g) with 6 mM TBA oleate surfactant. (h) SEM image of a Ttrz-DI ND/TBA dispersion deposited on a glass substrate. (i) Fluorescence microscope image of a Ttrz-DI ND/TBA oleate dispersion deposited on a glass substrate.”

2) We have unified all of the optical images as the reviewer suggested. However, scales on the SEM images have significantly different range than the optical microscope images. Therefore, we have kept the magnification used for the SEM images.

3) We agreed with the reviewer and repeated that experiment. New images are added in figure S8, which has been copied below:

“

Figure S8. Optical images and particle distribution graphs of the 4tBuMB nano-dots at (a, b) 0 mM, (c, d) 0.2 mM, (e, f) 0.8 mM, (g, h) 2 mM, (i, j) 4 mM, (k, l) 6 mM, (m, n) 8 mM, (o, p) 10 mM of TritonX-100 surfactant concentrations. The concentration of 4tBuMB was kept constant at 0.01 mM; (q) Table showing their mean, standard deviation, minimum and maximum sizes.”

4) We have provided particle size distribution graphs for all the optical microscope images (Figure S5, S6, S7, S8, S9, S10, S16, S17, S18) as the reviewer suggested, an example of which can be seen in our response to figure S8, above.

5) Figure S12 represents line profile analysis of an AFM image of Ttrz-DI nano-dots deposited on a silicon substrate from the nano-dots dispersion. The lines on the images passing through the nano-dots represent the profile of that line where the tip of the AFM cantilever scans the surface. Any particle on that line would be scanned by the cantilever tip and recorded as a peak in the profile, which gives information about its height and width. From the image, we have found that the nano-dots deposited on the substrate has a flattened pancake shaped structure with the height significantly smaller than the width on the silicon substrate. We have updated the figure quality and also included additional information about the figure in the caption.

“

Figure S12. Atomic force microscopy (AFM) line profile analysis of Ttrz-DI NDs. AFM topography

images (a, c, e, g, i, k) of a Ttrz-DI nano-dots deposited on silicon substrates and corresponding line profile analyses (b, d, f, h, j, l) were used to determine particle diameters.

Reviewer #3 (Remarks to the Author):

The authors reported the fluorescent organic nano dots prepared from the organic conjugated fluorescent molecules and surfactant. They investigated the effect of the ratio of surfactant (TBA and Triton X-100) on the fluorescence of nano dots. Both ionic and non-ionic surfactant have similar effect. They also put the fluorescent nano dots into the polymer matrix and demonstrated color conversion layer for convert the low wavelength to high wave length. However, the following concerns need to be addressed.

1. The novelty of this manuscript is open to discuss. This strategy has been developed for long period such as phase transfer technique. Because the concentration quenching effect, the report on the organic fluorescent nano dot is not frequently. But the AIE in copolymers have been investigated. Even in the this report, I believe, the authors need to control the amount of the organic fluorescent molecules.

[Our response] We thank the reviewer and agree with the comment. We also think that the method of producing fluorescent nanoparticles using molecular fluorophores with surfactants has been largely overlooked by the research community, likely due to the problems associated with molecular fluorophore NPs, such as concentration quenching. Only a handful of studies have characterized the use of surfactants with organic fluorophore particles, and generally have just used one surfactant concentration without any explanation of why this concentration was chosen. Although it has been clear that surfactants can benefit the properties of organic NPs, to our knowledge, no work has yet been published that provides a thorough investigation of how surfactants affect the properties of modern molecular fluorophores. Additionally, the incorporation of fluorescent organic nanoparticles in color conversion layers of light-emitting diodes is also very rare.

We've added the following statement on page 2 of the revised manuscript and added two references to address this issue:

“Due to concentration quenching^{20,21} effects observed in many fluorescent molecules, NPs based on molecular fluorophores have not been investigated as often as inorganic NPs, and a systematic, quantitative analysis of the effects of surfactants on organic NP properties such as yield, morphology and photo-physical properties, as well as the influence of surfactant properties, such as critical micelle concentration, is currently lacking.”

20. Lakowicz, J. R. Quenching of Fluorescence BT - Principles of Fluorescence Spectroscopy. in (ed. Lakowicz, J. R.) 257–301 (Springer US, 1983). doi:10.1007/978-1-4615-7658-7_9.

21. Chaudhuri, K. D. Concentration quenching of fluorescence in solutions. *Zeitschrift für Phys.* 154, 34–42 (1959).

2. The morphology controls strongly depends on the concentration of surfactant, To some concentration of surfactant, the micelles will form specific morphology. That is why 6 mM give the best morphology. Thus, the ratio of the fluorescent molecules to surfactant decide the fluorescent intensity due to the concentration of fluorescent molecules in surfactant. It is common issue, even without prove.

[Our response] We thank the reviewer for the comment. We agree that the morphology of the nanoparticles is strongly dependent on the surfactant concentration. Nanoparticles start to form when the surfactant concentration reaches critical micelle concentration (CMC). We found large particle sizes when the concentration of surfactant was below CMC and the size became smaller with increased surfactant concentration. The CMC value for the surfactants we used is in the range of 0.2~0.9 mM

however, at these concentrations the average size of the nanoparticles was much larger for our desired application and uniformity was also an issue. Therefore, we kept on increasing the concentration until the desired size and uniformity were optimized.

3. The yield of nano dots depends on the properties and CMC of surfactant. fluorescent molecules has minor effect.

[Our response] We thank the reviewer for the comment. We agree that the yield of nano-dots depends mostly on the CMC of the surfactant. We have added the following line in the main text (page 9) as suggested by the reviewer.

“The yield of nano-dots was largely dependent on the concentration of surfactants relative to the CMC of the surfactants rather than the structure of the fluorophore.”

4. I do not understand why the author define color conversion efficiency (CCE). What is the different between the PLQY and CCE?

[Our response] We thank the reviewer for the query about color conversion efficiency definition. The definition of CCE we provided was a simplified version of the following two articles.

“The color-conversion efficiency (CCE) refers to the proportion of QD light radiant power to the total radiant power.”-

-Li, J.; Tang, Y.; Li, Z.; Ding, X.; Yu, S.; Yu, B. Improvement in Color-Conversion Efficiency and Stability for Quantum-Dot-Based Light-Emitting Diodes Using a Blue Anti-Transmission Film. *Nanomaterials* 2018, 8, 508. <https://doi.org/10.3390/nano8070508>

“This implies that the color-conversion efficiency (η_c) can be calculated from ratio of the blue emission area for the LEDs without QDs deposition to the red emission area for the LEDs with QDs,”

-Park, M.J., Choi, K.J. & Kwak, J.S. Enhanced color-conversion efficiency between colloidal quantum dot-phosphors and nitride LEDs by using nano-patterned p-GaN. *J Electroceram* **33**, 2–6 (2014). <https://doi.org/10.1007/s10832-014-9892-6>

The simplified definition of CCE for a film would be the ability of the film to convert the light emitted by a LED to a certain wavelength depending on the properties of that film.

We've added the following statement in the description of Figure S19 of the supporting information to clarify how CCE was defined in this work:

“Color conversion efficiency is the ability of a color conversion layer to convert the incident light emitted by a LED to a certain wavelength depending on the properties of that layer.”

5. The author claimed the triton X-100 film in Figure 2e and 2i, it is hard to say film. it may just a layer of micellar particles stack on the substrate.

[Our response] We thank the reviewer for the comment and agree that these may not be continuous “films” in the conventional sense. We have replaced the word “film” by “deposited on glass substrate” to address this issue in the caption of Figure 2:

“**Figure 2.** Photoluminescence spectra of solution, film and nano-dot dispersion of Ttrz-DI with (a) Triton X-100, (b) TBA oleate. Optical microscope (OM) images of a Ttrz-DI dispersion deposited on a glass substrate synthesized (c) without surfactant and (d) with 6 mM Triton X-100 surfactant. (e) SEM image of a Ttrz-DI ND/Triton X-100 dispersion deposited on a glass substrate. (f) Fluorescence microscope (FM) image of a Ttrz-DI ND/Triton X-100 dispersion deposited on a glass substrate. Optical microscope images of a Ttrz-DI dispersion deposited on glass a substrate synthesized (g) without

surfactant (h) and with 6 mM TBA oleate surfactant. (i) SEM image of a Ttrz-DI ND/TBA dispersion deposited on a glass substrate. (j) Fluorescence microscope image of a Ttrz-DI ND/TBA oleate dispersion deposited on a glass substrate.”

6. color purity can be obtained from the CIE coordinates. it is not from FWHM.

[Our response] We thank the reviewer for the comment on color purity. We also agree with the reviewer that by definition color purity is obtained from CIE coordinates. However, FWHM is closely related to color purity and the location of the material on the CIE coordinates. Pure colors which occur on the outer edge of CIE space correspond to monochromatic light (FWHM = 0), and coordinates move towards the center of the CIE diagram (white light) as emission spectra become broader and the FWHM increases. Thus, we feel that FWHM is a useful figure of merit when color purity is concerned. We have rephrased the objectionable sentence in the main text (page 12) as quoted below:

“Figure S20-a compares the FWHM values of all the fluorophores used in this study where, CzDABNA showed the narrowest FWHM of 21.1 nm and Ttrz-DI showed the widest FWHM of 87.4 nm along with 4CzIPN (74.4 nm), 4tBuMB (36.8 nm) and TNAP (35.4 nm) having intermediate FWHM values. A color gamut (CIE 1931) was computed using the PL spectra of the nano-dots in Figure S20-b which shows that organic nano-dot synthesized by this method cover a wide gamut in the RGB color space comparable to the present day commercial NTSC standard color gamut used in displays.⁵¹ Boron containing nano-dots like TNAP and CzDABNA with narrow FWHM and high color purity (obtained from CIE coordinates) could find merit as potential candidates in display application since their color gamut occupies 105.96 % area compared to NTSC color space in the CIE coordinates. Whereas, nano-dots of TADF materials like Ttrz-DI and 4CzIPN with broader FWHM shows 71.05% and 76.91% area compared to NTSC color space, would be applicable as color conversion layers in white LEDs for future lighting purposes. Table 1 summarizes the PLQY, CCE, and related properties for different dispersions and films.”

Thanks,

Prof. Jang Hyuk Kwon

REVIEWER COMMENTS

Reviewer #1 (Remarks to the Author):

The authors tried their best to answer the reviewers' questions and to improve the quality of the manuscript, I believe it could be accepted at the current form.

Reviewer #2 (Remarks to the Author):

In this paper, ionic and non-ionic surfactant were used to synthesize nano-dots of organic fluorescent materials, the effect of surfactant on their properties was studied in detail, and the nano-dots were used as color conversion layer. The revised draft contains detailed replies and supplements to the questions, and it can be accepted for publishing in Nature Communication.

Reviewer #3 (Remarks to the Author):

The authors have addressed what I am concerned about. Agree to accept.